# A System for Assessing Dual Action Modulators of Glycine Transporters and Glycine Receptors

**DOI:** 10.3390/biom10121618

**Published:** 2020-11-30

**Authors:** Diba Sheipouri, Casey I. Gallagher, Susan Shimmon, Tristan Rawling, Robert J. Vandenberg

**Affiliations:** 1School of Medical Sciences, University of Sydney, Sydney, NSW 2006, Australia; dshe4741@uni.sydney.edu.au (D.S.); cgal2547@uni.sydney.edu.au (C.I.G.); 2School of Mathematical and Physical Sciences, University of Technology Sydney, Sydney, NSW 2007, Australia; susan.shimmon@uts.edu.au (S.S.); Tristan.Rawling@uts.edu.au (T.R.)

**Keywords:** glycine transporter, glycine receptor, analgesics, lipids

## Abstract

Reduced inhibitory glycinergic neurotransmission is implicated in a number of neurological conditions such as neuropathic pain, schizophrenia, epilepsy and hyperekplexia. Restoring glycinergic signalling may be an effective method of treating these pathologies. Glycine transporters (GlyTs) control synaptic and extra-synaptic glycine concentrations and slowing the reuptake of glycine using specific GlyT inhibitors will increase glycine extracellular concentrations and increase glycine receptor (GlyR) activation. Glycinergic neurotransmission can also be improved through positive allosteric modulation (PAM) of GlyRs. Despite efforts to manipulate this synapse, no therapeutics currently target it. We propose that dual action modulators of both GlyTs and GlyRs may show greater therapeutic potential than those targeting individual proteins. To show this, we have characterized a co-expression system in *Xenopus laevis* oocytes consisting of GlyT1 or GlyT2 co-expressed with GlyRα_1_. We use two electrode voltage clamp recording techniques to measure the impact of GlyTs on GlyRs and the effects of modulators of these proteins. We show that increases in GlyT density in close proximity to GlyRs diminish receptor currents. Reductions in GlyR mediated currents are not observed when non-transportable GlyR agonists are applied or when Na^+^ is not available. GlyTs reduce glycine concentrations across different concentration ranges, corresponding with their ion-coupling stoichiometry, and full receptor currents can be restored when GlyTs are blocked with selective inhibitors. We show that partial inhibition of GlyT2 and modest GlyRα_1_ potentiation using a dual action compound, is as useful in restoring GlyR currents as a full and potent single target GlyT2 inhibitor or single target GlyRα_1_ PAM. The co-expression system developed in this study will provide a robust means for assessing the likely impact of GlyR PAMs and GlyT inhibitors on glycine neurotransmission.

## 1. Introduction

Glycine is an inhibitory neurotransmitter in the central nervous system. Presynaptic release of glycine from inhibitory neurons activates strychnine-sensitive glycine receptors (GlyRs), causing an influx of Cl^−^ to hyperpolarize postsynaptic neurons [1]. There are four known GlyR α subunits (α_1_-α_4_), and one β subunit, which can assemble as α homopentamers, or as heteropentamers in a 2α:3β or 3α:2β arrangement [2]. Glycine transporters (GlyTs) control extracellular concentrations of glycine and influence the dynamics of glycinergic signaling [3]. Two subtypes have been identified in humans, GlyT1 and GlyT2 [4]. In addition to clearing glycine from the synapse, GlyT2 serves to recycle glycine for accumulation in presynaptic terminals of glycinergic neurons and subsequent exocytosis [5]. Glycine transport by GlyT2 is coupled to three Na^+^ and one Cl^−^, which allows it to maintain a high presynaptic glycine concentration of 20 mM which is required for efficient synaptic vesicle loading [3,5,6,7]. GlyT1 is expressed by glia surrounding both inhibitory glycinergic, and excitatory glutamatergic neurons, where its main role is the rapid reuptake of synaptic and extra-synaptic glycine [8,9,10] to terminate neurotransmission and regulate spill-over to glutamatergic synapses [11]. Glycine transport by GlyT1 is coupled to only 2 Na^+^ and 1 Cl^−^, which limits the maximal intracellular glycine concentration to approximately 2mM and also allows reverse transport to occur leading to release of glycine [3].

A range of neurological disorders arise from dysfunction in glycine receptors and transporters resulting in altered inhibitory transmission, which highlights the need to characterise the pharmacological modulation of the key proteins involved in glycinergic transmission. Although pathological changes in glycinergic signalling occur through various mechanisms, inhibition of the high affinity reuptake mechanisms of GlyTs or positive allosteric modulation of GlyRs have the capacity to increase glycinergic inhibitory tone and restore inhibitory signalling.

Despite a lot of effort to develop highly potent and selective GlyT inhibitors and GlyR potentiators, none of the compounds targeting these proteins have successfully made it through clinical trials as therapeutics. A number of reversible and potent bioactive lipid inhibitors of GlyT2 developed in our lab were recently found to also modulate GlyRs, which suggests that these lipids may have utility as dual modulators of these two key glycinergic proteins. Instead of aiming to develop compounds with high potency and selectivity for a single target, multi-target drug actions are aimed at a number of associated targets, often with reduced potency at each individual target, while still maintaining therapeutic efficacy [12,13,14]. Targeting multiple sites of the glycinergic synapse may therefore be a reasonable means to achieving greater improvements in glycinergic neurotransmission than targeting single proteins. In this study we have co-expressed GlyTs and GlyRs in *Xenopus laevis* oocytes to compare the pharmacological impact of single and dual acting compounds.

The activity of the bioactive lipids that inhibit GlyT2 and are also positive allosteric modulators of GlyR have been characterised separately [15,16,17,18]. When optimising the overall modulation of glycine neurotransmission, it would be useful to study how quantitative manipulation of GlyT activity directly impacts on GlyR activity. In the case of dual action modulators, their action at GlyTs may have direct consequences on the degree of potentiation exhibited at GlyRs. Here, we show that GlyTs can reduce the glycine concentration sensed at the membrane by GlyRs and alter the efficacy of receptor activation, establishing a reproducible and rapid system for quantifying the effects of GlyT inhibitors on GlyRs and for assessing the efficacy of dual action modulators.

Our group has developed a library of amino acids conjugated to lipid tails, which can be classified into three broad groups: (1) Potent, selective GlyT2 inhibitors that have minimal activity at GlyRα_1_, (2) potent positive allosteric modulators (PAMs) of GlyRα_1_ that have minimal activity at GlyT2 and (3) dual action lipids that are PAMs of GlyRα_1_ and inhibitors of GlyT2. From each of these three groups, the modulatory action of one lipid was tested in the GlyRα_1_/GlyT2 co-expression system. Here we show that a dual action lipid, N-oleoyl-glycine, targeting GlyRα_1_ and GlyT2 with relatively low affinity and efficacy is just as useful as single target lipids and suggest that modulating the glycinergic synapse to improve inhibitory neurotransmission without severe side-effect profiles may benefit from a multi-target approach.

## 2. Materials and Methods

### 2.1. Materials

C18-*cis*-ω9-glycine (or NOGly) was obtained from Sapphire Bioscience (Redfern, NSW, Australia), ALX-5407 and ORG-25543 were obtained from Tocris Bioscience (Victoria, Australia). All other *N*-acyl amino acids were synthesised as previously described. Ten mg mL^−1^ stock solutions of *N*-acyl amino acids, ALX-5407 or ORG-25543 were dissolved in DMSO and applied at a final concentration of 1 µM. Final solutions contained 0.0025% DMSO, a concentration which had no effect on transporter or receptor function.

### 2.2. Wild Type (WT) and Mutant RNA Transcription

Human GlyT1b and GlyT2a WT cDNA were subcloned into the plasmid oocyte transcription vector (pOTV) and human GlyRα_1_, GlyRα_3_ and GlyRβ into pGEMHE. GlyR cDNA pGEMHE vectors were provided by Mary Collins at the University of Sydney (Sydney, Australia). Single point mutations were introduced into GlyRα_1_ using the Q5 site-directed mutagenesis kit (New England Biolabs (Genesearch), Arundel, Australia) with oligonucleotide primers (Merck, Sydney, Australia) containing the desired mutations. The amplified cDNA/pOTV or cDNA/pGEMHE product was then transformed in *E. coli* cells, and subsequently purified using the PureLink Quick Plasmid Miniprep Kit (Invitrogen by Life Technologies, Löhne, Germany), and sequenced by the Australian Genome Research Facility (Sydney, Australia). The purified plasmid DNA was linearised via the restriction enzyme, *SpeI* (New England Biolabs (Genesearch), Arundel, Australia) for GlyT1b and GlyT2a (henceforth referred to as GlyT1 and GlyT2 respectively) and *NheI* (New England Biolabs (Genesearch)) for GlyRα_1,_ GlyRα_3_ and GlyRβ. Complementary RNAs were synthesised using the mMESAGE mMACHINE T7 kit (Ambion, Austin, TX, USA).

### 2.3. Oocyte Preparation and Injection

All work involving the use of animals was performed in accordance with the *Australian Code of Practice for the Care and Use of Animals for Scientific Purposes* and approved by the University of Sydney Animal Ethics Committee (Approval numbers 2016/970 and 2020/1704). (*Xenopus laevis* frogs (NASCO, Fort Atkinson, WI, USA) were anesthetised with 0.17% (*w*/*v*) 3-aminobenzoic acid ethyl ester and had an ovarian lobe removed via an incision in the abdomen. Stage V oocytes were isolated from the lobe via digestion with 2 mg/mL^−1^ collagenase A (Boehringer, Mannheim, Germany) at 26 °C for 1 h. 2 ng of cRNA encoding GlyT or GlyR subtypes were injected into each oocyte cytoplasm when transporters or receptors were studied individually. In GlyR/GlyT co-expressed cells, 2 ng of cRNA encoding GlyR and 6 or 20 ng of GlyT1 or GlyT2 encoding cRNA was injected into each cell (Drummond Nanoinject, Drummond Scientific Co., Broomall, PA, USA). Where GlyRα_1_β and GlyTs were co-expressed, a 1:5 ratio of GlyRα_1_ (2 ng) and GlyRβ (10 ng) cRNA was injected into single cells, as this ratio was found to be sufficient for formation of GlyR heteromers as judged by reduced sensitivity to pictrotoxin compared to GlyRα_1_ homomers. The oocytes were then stored in frog Ringer’s solution (96 mM NaCl, 2 mM KCl, 1 mM MgCl_2_, 1.8 mM CaCl_2_, 5 mM HEPES, pH 7.5) which was supplemented with 2.5 mM sodium pyruvate, 0.5 mM theophylline, 50 μg/mL gentamicin and 100 μM mL^−1^ tetracycline. The oocytes were stored at 18 °C for 3–5 days, until transporter and receptor expression were adequate for measurement using the two-electrode voltage clamp technique.

### 2.4. Two-Electrode Voltage Clamp Electrophysiology

GlyT1, GlyT2, GlyRα_1,_ GlyRα_3,_ GlyRα_1_β and GlyRα_3_β are electrogenic, allowing activation to be measured via the two-electrode voltage clamp technique. Oocytes were voltage clamped at −60 mV, and whole-cell currents generated by the substrate were recorded with a Geneclamp 500 amplifier (Axon Instruments, Foster City, CA, USA), digitised by a Powerlab 2/20 chart recorder (ADInstruments, Sydney, Australia). LabChart version 8 software (ADInstruments, Sydney, Australia) was used to visualise and process current traces. Recordings were performed in frog Ringer’s solution, except for low Na^+^ experiments where Na^+^ was replaced with choline. Oocytes were placed in an oval-shaped bath with a volume of 0.3 mL, with laminar flow around the oocyte at a rate of 12 mL min^−1^ under gravity feed.

### 2.5. Substrate and Agonist Concentration Responses

Varying concentrations of substrate or agonist were applied to cells in ND96. All glycine dose responses in the presence of bioactive lipids were performed using a 1 µM concentration of the lipid. Lipids were applied to the cell for 5 min prior to co-applying the lipid and glycine. Peak currents were measured and ND96 superfused the solution for a minimum of 10 min after each concentration of glycine was co-applied, to minimise GlyR desensitization. It was not possible to measure dose-responses in the presence and absence of lipid from the same cell, because recordings from a single cell were not viable for the 3+ h which are required to perform two dose-responses. Therefore, to minimise discrepancies, comparison between glycine dose responses in the presence and absence of lipid were made between cells recorded on the same day.

Current (I) as a function of glycine concentration was fitted by a least squares analysis to a derivative of the Hill equation:I/I_max_ = [substrate]*^n^*/([substrate]*^n^* + EC_50_^*n*^)
where I represents current (nA), I_max_ is the maximal current produced, EC_50_ is the concentration of substrate necessary to achieve half of the maximum response, and *n* is the Hill co-efficient. EC_50_ values are presented as mean and 95% confidence interval and Hill coefficients (*n*_H_) are presented as mean ± SEM.

### 2.6. Stop-Flow Recording

Flow of the solution was stopped until the current plateaued, by manually closing a three-way valve on one end of the bath such that the solution in the bath was static and is referred to henceforth as ‘stop-flow’.

### 2.7. Data Analysis

Data was analysed using GraphPad Prism 7 (version 7.02, GraphPad Software, San Diego, CA, USA). *n* refers to biological samples, not technical replicates. Data is normalised because the amplitude of currents vary significantly between cells and log transformations of data were undertaken to generate Gaussian-distributed datasets. Comparisons of two treatments in the same group were made using a two-tailed paired *t*-test and comparisons of two treatments in different groups were made using a two-tailed unpaired *t*-test. When multiple comparisons were tested, ANOVA with Dunnett’s multiple comparisons *post hoc* test was used. Significance threshold was set at * *p* < 0.05. Post hoc tests were only used if ANOVA reached significance.

## 3. Results

### 3.1. Glycine-Gated GlyR Peak Currents and Stopped-Flow Currents Are Reduced in Co-expressed Oocytes

*Xenopus laevis* oocytes were injected with complementary RNAs (cRNAs) encoding various combinations of GlyTs and GlyRs to generate a system where the effects of transporter function on receptor activity could be measured. Glycine-gated receptor currents were measured in oocytes expressing GlyRα_1_, GlyRα_1_/GlyT1 or GlyRα_1_/GlyT2. cRNA encoding the proteins were injected into oocytes at a ratio of 1 ng GlyR to 10 ng GlyT. Oocytes were voltage-clamped at −60 mV and superfused with solutions at 12 mL min^−1^ which generated large inward currents (Figure 1, I_flow_). In oocytes expressing only GlyRα_1_, currents generated by 10 µM glycine did not significantly change when the solution superfusing the oocyte was stopped. However, stopping the flow of solution superfusing oocytes, expressing GlyRα_1_ and either GlyT1 or GlyT2, resulted in a significant reduction in the current amplitude mediated by GlyRα_1_ (Figure 1, I_stop_). Following resumption of flow, currents rapidly returned to initial values. Both GlyTs also reduced peak current values in response to the same concentration of glycine (note scales, Table 1). These data suggest that expression of GlyT1 or GlyT2 modulates the activity of GlyRα_1_ when co-expressed in the same cell. In the following experiments, the influence of GlyT1 and GlyT2 on GlyR activation is further validated and characterised, and it is shown that GlyRs can be used as a sensor for concentration gradients of glycine created by the GlyTs.

As both GlyT1 and GlyT2 are electrogenic, glycine transport will contribute to the overall current, constituting noise in this system. Glycine dose-responses for GlyRs or GlyTs expressed alone show that the maximum values of currents elicited by GlyRs are in the thousands of nA, whereas the GlyTs elicit max currents in the hundreds of nA (Appendix A). Since GlyT currents are an order of magnitude smaller than GlyR currents, noise from GlyT currents are not expected to significantly distort GlyR peak current measurements. To verify the contribution of GlyTs to overall currents measured in co-expressed cells, glycine-gated currents were recorded in the absence and presence of a saturating concentration (1 µM) of the GlyR antagonist, strychnine. GlyRα_1_ sensitivity to strychnine is not affected by co-expression with GlyTs (Appendix A, IC_50_ and 95% confidence interval for GlyRα_1_: 8.23 (4.92 to 14.33), GlyRα_1_/GlyT1: 5.03 (3.94 to 6.53), GlyRα_1_/GlyT2: 3.25 (2.18 to 5.62) nM, significance tested using a one-way ANOVA and Dunnett’s post-hoc test). Cells expressing GlyTs alone also display stop-flow reductions in current (Appendix A), and from these measurements the contribution of GlyT currents to both fast-flow and stop-flow currents in co-expressed cells were estimated. At 10 µM glycine, GlyT1 and GlyT2 contributed 53.2 ± 6.9% and 16.6 ± 3.7% of the overall peak currents in GlyRα_1_/GlyT1 and GlyRα_1_/GlyT2 respectively. For the stopped flow currents, GlyT1 and GlyT2 contributed 61.5 ± 13.9% and 16.2 ± 2.1% for GlyRα_1_/GlyT1 and GlyRα_1_/GlyT2 respectively. At 30 µM glycine, which is close to the EC_50_ of both GlyTs and GlyRs, GlyT1 and GlyT2 contributed 14.4 ± 1.8% and 1.7 ± 0.4% to the overall peak current in GlyRα_1_/GlyT1 and GlyRα_1_/GlyT2 expressing cells respectively. For the stopped flow currents measured at 30 µM glycine, GlyT1 and GlyT2 contributed 5.2 ± 1.3% and 1.6 ± 0.4% of the currents for GlyRα_1_/GlyT1 and GlyRα_1_/GlyT2 respectively (Appendix A, mean ± SEM, *n* ≥ 5). These results suggest that both peak, and stop-flow currents shown in Figure 1 are mixed GlyR and GlyT currents and that the proportion of the two components of the current differ at different glycine concentrations.

### 3.2. Stop-Flow and Fast-Flow Reduction of Glycine Gated Currents Are Concentration Dependent

The reduction in the current amplitude in stop-flow conditions was then determined for a range of glycine concentrations. The ratio of current amplitude I_stop_/I_flow_, was found to be dependent on the concentration of glycine in the superfusing solution (Figure 2A,B,D,E, Table 2). Currents recorded during fast-flow and stop-flow conditions in the same cell also show a significant shift in the dose-response to the right and increase in associated glycine EC_50_ for in GlyRα_1_/GlyT1 or GlyRα_1_/GlyT2 expressing cells compared to GlyRα_1_ (Figure 2C,F). This suggests that GlyTs create diffusion-limited concentration gradients of glycine which are greatest in magnitude directly adjacent to the membrane.

### 3.3. Glycine Transporter Density Affects Degree of GlyR Modulation and Is Dependent on the Presence of a GlyT Transportable Substrate

Although the density of GlyTs throughout the CNS is yet to be established, alterations in the density of neurotransmitter transporters are known to shape signalling dynamics at inhibitory synapses [19]. Furthermore, increasing synaptic inhibitory neurotransmission as a therapeutic strategy to treat neurological and neuropsychiatric disorders are of considerable pharmacologic interest [10,20,21,22], and inhibition of GlyT mediated re-uptake of glycine is proposed as a rational way of achieving this. The equilibrium glycine concentration maintained by GlyT1 and GlyT2 is dependent on its ion flux coupling, whereas the rate at which equilibrium is achieved is dependent on the kinetics (turnover) and the number of available transporters. Modulation of GlyTs by partial and/or non-competitive inhibitors are expected to reduce the overall number of GlyTs available for glycine uptake, slowing the rate of glycine clearance without affecting the equilibrium concentration [3,20]. The subsequent increase in extracellularly available glycine is anticipated to prolong the time course of inhibitory synaptic transmission mediated by hyperpolarization of GlyRs in their vicinity.

The same amount of GlyRα_1_ encoding cRNA was injected into each cell (2 ng), regardless of whether GlyT encoding cRNA was co-injected, or in what ratio. To ensure the increased stop-flow uptake of glycine and the increased GlyRα_1_ glycine EC_50_ in cells co-expressed with GlyTs was indeed due to increased surface expression of GlyT and not decreased expression of GlyRα_1_, GlyRα_1_ and GlyT2 were tagged with mCherry and GFP fluorescent proteins respectively and fluorescence intensity, as a measure of cell surface protein expression, was determined by confocal microscopy. Functional analysis of GlyT2-GFP and GlyRα_1_-mCherry tagged proteins showed that they display very similar glycine sensitivities to their untagged counterparts (Appendix A).

To determine if it is possible to study the effects of transporter number on GlyRα_1_ stop-flow and fast-flow activation in this co-expression system, GlyRα_1_ cRNA was co-injected with GlyT1 or GlyT2 cRNAs in increasing ratios and the I_stop_/I_flow_ ratio and glycine EC_50_s were determined (Figure 3). For GlyRα_1_/GlyT1 (Figure 3A,B) ratios greater than 1:5 did not cause any further increase in the I_stop_/I_flow_ ratio (1:3 = 43.0 ± 10.1, 1:5 = 60.1 ± 6.8, 1:10 = 62.2 ± 2.9 and 1:20 = 65.9 ± 6.8%, mean ± SEM, *n* = 5). In contrast, in GlyRα_1_/GlyT2 expressing cells (Figure 3D,E), ratios greater than 1:3 created a further increase in the I_stop_/I_flow_ ratio (1:3 = 36.1 ± 5.7, 1:5 = 68.9 ± 3.4, 1:10 = 78.4 ± 1.7 and 1:20 = 80.1 ± 1.5%, mean ± SEM, *n* = 5). Glycine concentration dependent currents measured in co-expressed cells at GlyR/GlyT ratios of 1:3 and 1:10 showed that generation of glycine concentration gradients under fast-flow conditions occurred more robustly with GlyT1 compared to GlyT2. The shift of the GlyRα_1_ fast-flow dose-response to the right was greater in GlyT1 compared to GlyT2 co-expressed oocytes at both 1:3 and 1:10 ratios. GlyT1 also appeared to be less sensitive to the effects of decreased amounts of injected cRNA, as there was a smaller difference in its glycine EC_50_ of 1:3 and 1:10 cRNA injected oocytes (Figure 3, Appendix A).

Since the functional properties of tagged proteins were very similar to untagged proteins, the cell surface expression of the tagged proteins was also expected to be reliably representative. Cells were injected at 1:3 and 1:10 GlyRα_1_-mCherry: GlyT2-GFP. Glycine dose-responses were measured on GlyRα_1_-mCherry and GlyT2-GFP expressing cells, and fluorescence intensity in the same cell of were measured by confocal microscopy (Appendix A). Glycine dose responses for the different ratios of injected cRNA showed very similar EC_50_s to untagged proteins injected with the same ratios (see Appendix A). Fluorescence intensity of GlyT2-GFP was significantly increased in the 1:10 ratio injected oocytes compared to the 1:3 ratio injected oocytes, whereas there was no significant difference between the mean fluorescence intensity of GlyRα_1_-mCherry between the two ratios (Appendix A). This verifies that decreases in GlyRα_1_ glycine sensitivity in co-expressed GlyRα_1_/GlyT2 oocytes are due to increased number of available GlyTs. β-Alanine and taurine are GlyRα_1_ agonists [23], but are not transportable substrate of either GlyT1 or GlyT2 [24,25,26]. β-alanine dose-responses were measured in cells expressing only GlyRα_1_ and GlyRα_1_ co-expressed with either GlyT1 or GlyT2 (Figure 4). There was no significant difference in GlyRα_1_ sensitivity to β-alanine between all dose-responses (EC_50_ and 95% confidence interval for GlyRα_1_: 28.3 (95% CI: 26.5 to 30.4) for GlyRα_1_/GlyT1: 28.0 (95% CI: 26.6 to 29.4) and for GlyRα_1_/GlyT2: 27.4 (95% CI: 25.1 to 29.8). Significance was tested using a one-way ANOVA, *p* < 0.05). These data confirm that the reduction of current in co-expressed oocytes is dependent on the presence of a GlyRα_1_ agonist which is also a transportable substrate of GlyT2 or GlyT1.

### 3.4. Stop-Flow and Fast-Flow Reduction of Current in GlyRα1 Is Reliant on Na^+^ Dependent GlyT Driving Force

Glycine transport by GlyT1 is coupled to the co-transport of 2 Na^+^ and 1 Cl^−^, which allows the transporter to move substrate across the membrane both in and out of a cell depending on the physiological conditions. For GlyT2, glycine transport is coupled to the co-transport of 3 Na^+^ and 1 Cl^−^, which provides it with a large concentrating capacity and ensures that transport is directed inward under most physiological conditions [3]. Fast-flow and stop-flow glycine-gated GlyRα_1_ currents were measured in buffers with decreased Na^+^ concentrations, using choline as a substitute cation to keep total cation concentration equal. Removing Na^+^ from the superfusing solution was expected to prevent GlyT uptake of glycine and hence prevent glycine concentration gradients forming near the membrane. Example traces from cells expressing GlyRα_1_/GlyT1 (Figure 5A) or GlyRα_1_/GlyT2 (Figure 5D) show reducing the Na^+^ concentration in the perfusate has different stop-flow and fast-flow effects on the two transporters. For oocytes expressing GlyRα_1_/GlyT1, reducing the Na^+^ concentration in the perfusate significantly increases the peak glycine-gated currents in response 10 µM glycine compared to 96 mM Na^+^ in the same cell (Figure 5B, Table 3). There was also an increase in peak currents in GlyRα_1_/GlyT2 cells (Figure 5E, Table 3), although this change was not as marked.

Reducing the Na^+^ concentration in the perfusate from 96 mM to 10 mM, reduces the extent of stop-flow current reduction to a significant degree for GlyT1 (Figure 5C, Table 3), which demonstrates that when the driving force for transport is reduced, the impact on stop-flow current reductions is reduced. For GlyRα1/GlyT1 co-expressed cells, in the presence of 1 mM Na^+^, stopping the flow caused potentiation of GlyRα1 current amplitudes (Figure 5A, Table 3). Under these conditions, 1 μM ALX-5407 prevents the stopped flow increase in current amplitude (Appendix A), which confirms that the increased current is due to reverse glycine transport. In GlyRα_1_/GlyT2, stop-flow reduction in current amplitude was entirely prevented in 10 mM Na^+^ and in 1 mM Na^+^ buffers (Figure 5F, Table 3). These observations are in good agreement with electrophysiological measurements showing that the lower Na^+^ coupling stoichiometry of GlyT1 allows reverse transport of glycine out of the cell [3,27], and confirms the hypothesis that both GlyTs modulate receptor activation through ion-coupled glycine uptake and efflux. The lack of apparent reverse transport mediated by GlyT2 with 1 mM Na^+^, despite an outward directed Na^+^ gradient [5] suggests that the rate of reverse transport by GlyT2 is slower than that of GlyT1 such that there is no apparent effect on extracellular glycine concentrations in the time frame of the measurements in these experiments.

Similar glycine concentration-response curves were measured using oocytes expressing GlyRα_1_β, GlyRα_3_ and GlyRα_3_β alone or co-expressed with either GlyT1 or GlyT2 to determine the extent of the effect of GlyTs on different GlyR subtypes (Figure 6, Table 4).

GlyTs effectively create concentration gradients of glycine in the vicinity of all receptor subtypes, but it should be noted that the shift in the glycine concentration response curve was greater for GlyRα_1_ compared to other GlyR subtypes. This is a consequence of the EC_50_ of GlyRα_1_ being in closer alignment with the concentrations at which the GlyTs are most effective in clearing glycine. For GlyRα_3_, the EC_50_ is significantly higher and further away from the optimal range for clearance by the GlyTs.

### 3.5. Estimation of Glycine Sensed at the Membrane by GlyRs in Cells Co-Expressing GlyTs

The reductions in glycine gated currents mediated by GlyR under both stopped flow and fast flow conditions suggests that it is possible to quantify the glycine concentration sensed at the membrane when the GlyTs are co-expressed. The concentration of glycine sensed at the membrane by GlyRs in the presence of the GlyTs was estimated using the Hill equation. Using the mean EC_50_ and Hill coefficients (*n*) of glycine concentration-response curves from oocytes expressing only GlyRs, I_max_ and I values from GlyR/GlyT co-expressed cells were substituted into the reversed Hill equation to solve for [Gly]_m_ [28]. Figure 7 shows the estimate of glycine concentrations at the membrane in co-expressed cells for bath concentrations of glycine from 5–300 μM. Concentrations sensed at the membrane in co-expressed cells (dark blue lines) are lower than those applied in the bath (indicated by the light blue line for GlyRα_1_ alone).
(1)$$\begin{array}{ccc} \begin{array}{c} I=\frac{Imax}{1+{\left(\sqrt{\frac{EC_{50}}{{\left[Gly\right]}_{b}}}\right)}^{n}} \end{array} & ⟶ & {\left[Gly\right]}_{m}=\frac{EC_{50}}{\sqrt[{n}]{\frac{I_{max}}{I}-1}} \end{array}$$

Equation (1): The Hill equation was reversed to solve for the glycine concentration sensed at the membrane by GlyRs, [Gly]_m_. [Gly]_b_ is concentration of glycine in bath applied in the perfusate.

### 3.6. Pharmacological Inhibition of GlyTs Reverses Stop-Flow and Fast-Flow Changes to GlyR Activation Profiles

GlyRα_1_ was co-expressed with GlyTs in a ratio of 1:10 and stop-flow reduction of glycine-gated GlyRα_1_ currents were determined in the absence and presence of GlyT inhibitors. ALX-5407 is a potent inhibitor of GlyT1, with an IC_50_ of 3 nM [29,30]. The I_stop_/I_flow_ value for 10 µM glycine applied to GlyRα_1_/GlyT1 cells is 0.39 ± 0.05 and with co-application of ALX-5407 this value increases to 0.97 ± 0.01 in the same cell (Figure 8A,B, Table 5, mean ± SEM, *n* ≥ 5). ORG-25543 is a potent inhibitor of GlyT2 with an IC_50_ of 16 nM [31,32]. The I_stop_/I_flow_ value for 10 µM glycine applied to GlyRα_1_/GlyT2 cells is 0.33 ± 0.04 and with co-application of ORG-25543 this value increases to 0.90 ± 0.04 in the same cell (Figure 8D,B,E, Table 5, mean ± SEM, *n* ≥ 5). This data shows co-application of a supra-maximal 1 µM concentration of the GlyT inhibitors, ALX-5407 or ORG-25543 with 10 µM glycine in GlyRα_1_/GlyT1 or GlyRα_1_/GlyT2 cells_,_ respectively, prevents the reduction in the I_stop_/I_flow_ value (close to 1, i.e., no stop-flow reduction) compared to application of glycine alone in the same cell. GlyRα_1_ sensitivity to glycine was also measured under fast-flow conditions was determined by measuring glycine concentration dependent currents in GlyRα_1_/GlyT co-expressed cells, in the presence and absence of supra-maximal concentrations of GlyT inhibitors (Figure 8C,F, Table 5). The fast-flow glycine EC_50_ value for GlyRα_1_/GlyT1 is 48.8 µM and with co-application of ALX-5407 this value decreases to 11.3 µM, which is not significantly different from the glycine EC_50_ value for GlyRα_1_ alone, 11.2 µM (Figure 8C, Table 5). The fast-flow glycine EC_50_ value for GlyRα_1_/GlyT2 is 41.6 µM and with co-application of ORG-25543 this value decreases to 13.2 µM, which again is not significantly different from the glycine EC_50_ value for GlyRα_1_ alone, 11.2 µM (Figure 8F, Table 5). The concentration gradients generated by GlyT1 or GlyT2 under fast-flow conditions, in response to a range of glycine concentrations, can therefore also be reversed by co-application of a supra-maximal concentration of 1 µM ALX-5407 or 1 µM ORG-25543 respectively. These data reiterate that GlyTs decrease the concentration of glycine at the membrane sensed by GlyRα_1_ and verify that pharmacological manipulation of GlyTs in this system can be assessed by subsequent changes in GlyRα_1_ activation profiles.

### 3.7. Co-Expression of GlyTs with GlyRs Changes Hill Co-Efficient Values

GlyRα_1_ channels form as pseudo-symmetrically arranged pentamers, with the agonist binding domains in clefts between adjacent subunits [33,34]. Hill co-efficients have been used to describe cooperativity in agonist activation of GlyRs and these values can be altered by modulators or mutations which cause direct or allosteric disruption to the agonist binding affinity. It is unlikely that incorporation of GlyTs change agonist binding affinity to GlyRα_1_ through a direct physical mechanism, because the increase in Hill co-efficient can be reversed when GlyTs are inhibited (Table 5). GlyT1 and GlyT2 have glycine EC_50_ values of about 35 µM and 30 µM (Figure 9) respectively and are saturable. This makes them most effective at creating glycine concentration gradients in the bath in asymmetrical regions of the GlyR glycine dose response curve, changing the slope of the curve and creating an apparent change in affinity.

The Hill co-efficient for glycine dose-responses in GlyRα_1_ is 2.2 ± 0.4 (Table 5) which agrees well with previous reports [35,36], however it has been reported as both higher [37] and lower [2,38] by others. Expression of both GlyT1 and GlyT2 increases the Hill co-efficient (*n*_H_) of GlyRα_1_, to 3.8 ± 0.2 and 3.2 ± 0.3 respectively, however this increase is only significant in the case of GlyT1. When GlyT inhibitors are co-applied, the Hill co-efficient decreases back to values which are not significantly different to those for GlyRα_1_ alone (2.0 ± 0.2 and 2.6 ± 0.4 respectively) (Table 5).

### 3.8. Novel, Bioactive Lipid Modulators of GlyRα1 and GlyT2

The effects of the GlyT2 specific inhibitor, C18-*cis*-*ω*9-L-methionine [17], the GlyR specific PAM, C18-*cis*-*ω*7-glycine [18] and the dual action modulator C18-*cis*-*ω*9-glycine [17,18] were assessed in co-expressed GlyRα_1_/GlyT2 cells. As the data previously collected on the activity of these lipids at GlyRα_1_ was measured using a 1 µM concentration [18] all the following experiments have also been conducted using a 1 µM concentration. For compounds with GlyT2 inhibitory activity, 1 µM should also provide near maximal levels of inhibition [15,17]. These GlyT2 targeting lipid compounds have negligible effect on GlyT1 activity [15,17], and therefore all subsequent experiments assessing pharmacologic modulation are performed on GlyRα_1_/GlyT2 cells.

C18-*cis*-*ω*9-L-methionine is a potent inhibitor of GlyT2, with an IC_50_ value of 29.2 nM and at 1 µM, inhibits glycine transport by 84.9 ± 0.07%. GlyRα_1_ modulation at 1 µM is limited, with C18-*cis*-*ω*9-L-methionine potentiating currents induced by an EC_5_ application of glycine by 2.2 ± 4.2% at 1 µM (Figure 10, Table 6). This lipid can therefore be used to assess the action of selective GlyT2 inhibitors on GlyRα_1_/GlyT2 activity. C18-*cis*-*ω*7-glycine inhibits GlyT2 by 10.5 ± 0.02% at 1 µM but is a potent positive allosteric modulator (PAM) of GlyRα_1_ potentiating the EC_5_ of glycine by 184 ± 41.3 % (Figure 10, Table 6). This lipid can therefore be used in this co-expression system to assess the action of selective GlyRα_1_ PAM on GlyRα_1_ activity in the vicinity of GlyT2. For context, other lipid modulators which induce analgesia by acting on GlyRα_1_, potentiate EC_5_ currents by ~400–800% when assessed in HEK-293 cells [39,40].

One μM C18-*cis*-*ω*9-glycine inhibits GlyT2 by 50.1 ± 0.06%, but it is also a moderately effective PAM of GlyRα_1_, with 1 µM potentiating EC_5_ glycine currents by 84.7 ± 14.0% (Figure 10, Table 6). With the GlyRα_1_/GlyT2 co-expression system, it is now possible to investigate the effects of lipids with dual actions at GlyT2 and GlyRα_1_.

Glycine dose-response curves were measured in cells expressing GlyRα_1_ or co-expressing GlyRα_1_/GlyT2. To minimise discrepancies between GlyT2 expression levels in different batches of cells, glycine dose-response curves in the presence of lipid were measured in GlyRα_1_ expressing and GlyRα_1_/ GlyT2 co-expressed cells on the same day as control dose-responses, using cells from the same batch. The efficacy of lipids in co-expressed cells is assessed by their ability to return sensitivity to glycine back to that for GlyRα_1_ alone, as with ORG-25543. In the absence of lipid, glycine EC_50_ values for GlyRα_1_/GlyT2 cells were significantly increased compared to GlyRα_1_ (Figure 11, Table 7), as previously demonstrated. Glycine dose-responses were then measured in the presence of 1 µM of each of these lipids, on cells expressing GlyRα_1_ or GlyRα_1_/GlyT2. There is no significant change in Hill co-efficient values between GlyRα_1_ expressing cells ± lipid or GlyRα_1_/GlyT2 expressing cells ± lipid (Table 7).

C18-*cis*-*ω*9-L-methionine is expected to decrease the glycine EC_50_ of GlyRα_1_/GlyT2 cells, as it prevents the uptake of glycine by GlyT2, and should behave in a similar manner to ORG-25543. As expected, co-application of C18-*cis*-*ω*9-L-methionine with glycine to GlyRα_1_ expressing cells causes no significant shifts to the glycine dose-response and associated EC_50_. However, for GlyRα_1_/GlyT2 expressing cells, C18-*cis*-*ω*9-L-methionine shifts the glycine dose-response to the left, with a corresponding reduction in the glycine EC_50_ from 27.9 to 17.4 µM, which is slightly greater than the glycine EC_50_ value for GlyRα_1_ alone, 14.8 µM (Figure 11A, Table 7).

C18-*cis*-*ω*7-glycine is expected to decrease the glycine EC_50_ of both GlyRα_1_ and GlyRα_1_/GlyT2 expressing cells. It has minimal activity at GlyT2 and therefore, it’s expected that any lipid induced change in the apparent glycine sensitivity is due to stimulation of GlyRα_1_. As expected, co-application of C18-*cis*-*ω*7-glycine with glycine shifts the dose responses to the left with corresponding reductions in the glycine EC_50_ in GlyRα_1_ cells (EC_50_ shift from 13.2 to 9.3 and 13.9 to 9.2 µM respectively, Figure 11B, Table 7). In GlyRα_1_/GlyT2 cells, C18-*cis*-*ω*7-glycine shifts glycine dose responses to the left with corresponding reductions in the glycine EC_50_ (EC_50_ shift from 27.3 to 21.7, Figure 11B, Table 7), which reflects the direct stimulation of GlyRα_1_.

In GlyRα_1_ expressing cells, the dual action lipid, C18-*cis*-*ω*9-glycine, is expected to shift the glycine dose-response curve to the left and decrease the associated glycine EC_50_. The potentiating activity measured at the glycine EC_5_ is relatively modest (Figure 10B, Table 6), so the effects are not likely to be as marked. In GlyRα_1_/GlyT2 expressing cells, C18-*cis*-*ω*9-glycine is expected to shift the glycine dose response curves for GlyRα_1_ and simultaneously limit the clearance of glycine from the system due to GlyT2. The combined effects should shift the overall glycine dose response curve to the left, with a decrease in associated glycine EC_50_. When C18-*cis*-*ω*9-glycine is applied in co-expressed cells, some uptake of glycine by GlyT2 is expected to still occur, and this will reduce the actual concentration of glycine reaching GlyRα_1_.

In the presence of C18-*cis*-*ω*9-glycine, the glycine dose response curve for GlyRα_1_ expressing cells is shifted to the left (EC_50_ changes from 15.7 to 13.4 µM) (Figure 11C, Table 7), however this change is not statistically significant. From the glycine dose-response curve, the most marked changes occur at low glycine concentrations (5–10 µM), but the overall change across all concentrations is not significant. In GlyRα_1_/GlyT2 expressing cells, C18-*cis*-*ω*9-glycine causes a leftwards shift of the glycine dose response curve and associated EC_50_ from 34.1 to 18.0 µM (Figure 11C, Table 7). This value is not significantly different from the glycine EC_50_ for GlyRα_1_ (15.7 µM) suggesting that the dual action of a partial GlyT2 inhibitor and a moderate GlyRα_1_ potentiator has the capacity to perform as well as a full GlyT2 inhibitor, ORG-25543 (*cf.*
Figure 8F). The shift in glycine EC_50_ for the dual action compound C18-*cis*-*ω*9-glycine is also very similar to that elicited by a very efficacious lipid GlyT2 inhibitor (*cf.*
Figure 4A and Figure 5A) or a very efficacious GlyRα_1_ potentiator (*cf.*
Figure 4B and Figure 7B). These data verify the utility of this co-expression system to study the effects of simultaneous modulation of two key proteins at the glycinergic synapse.

### 3.9. Estimation of the Apparent Glycine Concentration Sensed at the Membrane by GlyRα1 in GlyRα1/GlyT2 Co-Expressed Cells, in the Presence of Lipid Modulators

Using the EC_50_ and Hill co-efficient (*n*_H_) of glycine dose-response curves from cells expressing GlyRs only, I_max_ and I values from GlyR/GlyT co-expressed cells were substituted into the reversed Hill equation to solve for [Gly]_m_ (Figure 7) [28]. Using the same principal, the apparent concentration of glycine sensed at the membrane in GlyRα_1_/GlyT2 expressing cells in the presence of the lipid modulators can be calculated. For the cases where a lipid inhibitor of GlyT2 is used, the glycine concentration sensed at the membrane is restored back to the bath glycine concentration. In the cases where a GlyRα1 PAM or a dual action GlyT2 inhibitor/GlyRα1 PAM are used, the apparent glycine concentration is greater than the bath glycine concentration. This apparent anomaly is a measure of the greater sensitivity of the receptor to glycine caused by the lipid. Figure 12 shows the estimate of the apparent glycine concentrations at the membrane for bath concentrations of 5–100 μM glycine in the presence of the different classes of lipid modulators.

## 4. Discussion

The aim of this work was to develop and characterize a reproducible system that can be used to rapidly assess the impact of glycine transporters on the function of glycine receptors, and to assess the outcomes of dual pharmacological modulation of these two proteins. By co-expressing GlyRs and GlyTs in *Xenopus* oocytes, we have shown that GlyTs reduce GlyR mediated currents by reducing extracellular glycine concentrations in a transporter density dependent manner.

### 4.1. The Impact of GlyTs on GlyR Function

Both GlyT1 and GlyT2 decrease the current amplitudes under stop-flow conditions and fast flow conditions, and the magnitudes of the decreases in current are dependent on the concentrations of glycine applied in the bath. This indicates that GlyTs create glycine concentration gradients at the membrane which are limited by free diffusion of glycine from the bath. The formation of concentration gradients through active transporter uptake in the presence of an unstirred layer of solute adjacent to membranes has been reviewed [41], and discussed in the context of co-expression studies [28]. Briefly, the unstirred layer is a region adjacent to the membrane in which no mixing of the bulk solution occurs during laminar flow. Active transport by GlyTs expressed on the membrane surface accumulate glycine intracellularly, depleting the concentration of glycine on the extracellular side of the membrane. This forms a glycine concentration gradient in this unstirred layer, which is limited by passive diffusion from the stirred, bulk solution in the bath, to the membrane surface. With increasing glycine concentrations, stop-flow uptake by GlyTs could be diminished and eventually stopped, which is not surprising given that uptake of glycine by GlyTs is a saturable process. The steepness of this gradient is expected to be larger in circumstances where diffusion is limited by space, as in the neuronal synapse [41].

We investigated the effect of varying the amount of GlyTs on GlyR activity and demonstrated that the higher GlyT to GlyR ratios, the greater the reduction in receptor activity, particularly in the case of GlyT2. This shows that this co-expression system can be used to model the effects of transporter number on receptor activity. In cases where GlyT expression is decreased, or function is pharmacologically inhibited, the concentration gradients generated are expected to be smaller and the increase in EC_50_ is also expected to be smaller. The number of active GlyT proteins present at the presynaptic membrane and astrocytes in vivo is highly dynamic and dependent on various mechanisms including interaction with proteins, modulation by Ca^2+^, regulation by several intracellular trafficking pathways, interaction with lipid rafts, and purinergic signalling [42,43,44,45]. Under each of these conditions, the clearance rates for glycine and the impact on GlyR function would also change. Furthermore, GlyT1 and GlyT2 expression is regulated in opposite ways by some of these modulators, highlighting the interplay of these two transporters in excitatory and inhibitory neurotransmission. It would be of interest to apply this system to investigate the impact of plasticity of GlyT expression in vivo, particularly in pathological states such as hyperekplexia where mutations in GlyT2 are known to affect localisation and decrease expression levels [46]. Since transporter number can be modelled in this system, pharmacological inhibition of GlyTs, by a range of different types of inhibitors could be studied [20].

The glycine EC_50_ we have reported here (13.2 μM) for GlyRα_1_ is lower than reported by a number of studies using two-electrode voltage clamp electrophysiology to study this ion channel [47,48,49,50]. However, it has been previously shown that glycine EC_50_ values can be highly variable in oocytes, ranging from ~20–280 μM [35]. It was found that in cells with low maximal currents (~200 nA), the EC_50_ values are greatest and in cells with the greatest maximal currents (~10 000 nA) the EC_50_ values are smallest. Our maximal currents for GlyRα_1_ are consistently in the range of 1000′s of nA, which may explain why our EC_50_ values are in thelower end of this range. Furthermore, similar EC_50_ values of 24 [51] and 42 μM [52] have also been recently reported.

GlyRs that consist of α1β subunits are the predominant subtype found in the adult CNS [53]. We looked at the effects of GlyTs on glycine concentration-responses generated by these receptor subtypes. Incorporation of the β subunit did not significantly change the sensitivity of GlyRs to glycine, which is in good agreement with previous reports of heterologous expression in *Xenopus* oocytes [2,36]. However, in patch-clamp studies, peak current amplitudes mediated by α_1_β heteromers have a higher sensitivity to glycine than α_1_ homomers [54,55,56,57]. Fast peak current amplitudes cannot be obtained by the two-electrode voltage clamp technique using *Xenopus laevis* oocytes and could explain why we and others have found a similar sensitivity to glycine for heteromeric and homomeric GlyRs using this technique.

Both GlyT1 and GlyT2 can modulate concentrations of glycine in the vicinity of GlyRs, with most pronounced effects observed for glycine concentrations in the range of 1–100 μM. The degree of change in EC_50_ for glycine when the GlyTs were co-expressed with GlyRs correlates with the EC_50_ of the GlyR in the absence of GlyTs. For GlyRα_1_ receptors (EC_50_ 13.2 μM), GlyT1 and GlyT2 caused 3.75 and 3.15-fold increases in EC_50_, respectively (Table 4), whereas for GlyRα_3_β (EC_50_ 109.4), GlyT1 and GlyT2 caused only 2.00- and 2.05-fold increases in EC_50_, respectively. These differences can be attributed to the optimal concentration at which the GlyTs can clear glycine, and as the EC_50_ for GlyRα1 is closest to the optimal range of 1–100 μM glycine this receptor shows the marked changes in EC_50_. Although inhibiting GlyTs to improve glycinergic neurotransmission will increase glycine-gated currents at α_1_ and α_3_ containing receptors, the same degree of GlyT inhibition will likely increase the sensitivity of α_3_ containing receptors to a different degree compared to α_1_, as local increases in glycine concentration will affect different parts of the glycine dose-response curves.

Following presynaptic release of glycine, synaptic glycine rapidly reaches ~3 mM to ensure efficient GlyR activation [58]. However, in the co-expression system characterized in this study, GlyTs have greatest impact on GlyR activity in the 1–100 µM range, which is well below the peak glycine concentration that generates inhibitory post-synaptic currents (IPSCs) but is in the range of glycine concentrations that may spill over from glycinergic synapses to neighbouring synapses. Thus, this system is most likely to reflect the way that GlyTs can regulate glycine concentrations at extra-synaptic sites and regulate tonic GlyR activity [59]. In two studies of pharmacological manipulation of glycine neurotransmission in the dorsal horn of the spinal cord, it has been demonstrated that bioactive lipid inhibitors of GlyT2 have minimal effects on IPSCs but have very marked effects on tonic GlyR activity [59,60]. These inhibitors show marked analgesic effects in rat models of neuropathic and inflammatory pain and so it is conceivable that the analgesic effects are due to modulation of tonic GlyR activity. Therefore, this co-expression system could be a useful tool to study how these inhibitors may alter tonic GlyR activity.

### 4.2. Pharmacological Modulation of the GlyR/GlyT Co-Expression System

Inhibitors of GlyT2 have been shown to have analgesic actions in inflammatory and neuropathic pain models but have not been developed as therapeutics due to either poor pharmacokinetic, selectivity or reversibility profiles [61]. More recently, attention has turned to lipid inhibitors of GlyT2 based on the structure of *N*-arachidonylglycine, an endogenous lipid that is structurally related to anandamide [15,17,59,60,62,63], which show promising analgesic activity. GlyR potentiators have also shown promising results as novel analgesics in behavioural mouse models of neuropathic pain [64]. The model developed here could be applied to provide a more comprehensive understanding of how these various compounds modulate glycinergic transmission.

First, we demonstrated the utility of the co-expression system by confirming that the well characterized GlyT1 and GlyT2 inhibitors, ALX5407 and ORG25543, could reverse the effects on GlyR currents caused by co-expression of GlyT1 and GlyT2 respectively. A library of bioactive lipid inhibitors of GlyT2 have been developed [15,17] and more recently, some of these bioactive lipids have been shown to also act as positive allosteric modulators of GlyRs [18]. This opens the possibility of developing dual action compounds that have the potential to provide synergistic effects on glycinergic neurotransmission. The effects of a select group of bioactive lipid inhibitors were explored in the co-expression system developed here. The activities of different bioactive lipids were assessed based on their mode of action, i.e., a GlyT2 inhibitor, a GlyR PAM, and a dual action GlyT2 inhibitors/GlyR PAM. C18-*cis*-*ω*9-L-methionine is a selective GlyT2 inhibitor with minimal activity at GlyRα_1._ It has minimal effects on the glycine EC_50_ for GlyRα1 but reduces the EC_50_ for glycine activation of GlyRα_1_/GlyT2 cells which is consistent with its effects as a GlyT2 inhibitor. The effects were not as pronounced as observed for ORG25543 (Figure 11A), which was also expected, since it is not a full inhibitor at 1 μM (84.9% inhibition). This observation further confirms the utility of this co-expression system to study the pharmacological effects of GlyT2 inhibitors in modulating GlyT2/GlyR activity. The single target GlyRα_1_ PAM, C18-*cis*-*ω*7-glycine, decreases the glycine EC_50_ in cells expressing GlyRα_1_ alone and also decreases the glycine EC_50_ for the GlyRα1/GlyT2 cells. The dual action compound, C18-*cis*-*ω*9-glycine, shows moderate efficacy as a GlyT2 inhibitor and GlyRα_1_ PAM. Whilst it is a less potent inhibitor of GlyT2 than ORG-25543, it appears that its additional action on GlyRα_1_ compensates such that the net response is similar to that of ORG-25543. C18-*cis*-*ω*9-glycine decreases the glycine EC_50_ values and restores GlyRα_1_ sensitivity to glycine in co-expressed cells to a greater degree than the moderately efficacious single target GlyRα_1_ PAM, C18-*cis*-*ω*7-glycine [17,18]. This data shows that a dual action lipid with moderate activity at GlyRα_1_ and GlyT2 can be just as efficacious as a full GlyT2 inhibitor, or very potent lipid GlyT2 inhibitors and GlyRα_1_ PAMs.

Whilst the dual action compounds show similar overall effects to potent and selective GlyT2 inhibitors, there are some subtle differences which can be explained by their actions at the two sites. It has previously been shown that C18-*cis*-*ω*9-glycine decreases the glycine EC_50_ and also increases the I_max_ of GlyRα_1_ in a dose-dependent manner [18], which could explain the similar overall efficacy as ORG-25543. This observation is also reflected in the estimate of the apparent bath concentration of glycine (Figure 7, Figure 12). Thus, the additional actions of C18-*cis*-*ω*9-glycine in decreasing the EC_50_ for glycine at GlyRα_1_ and increasing the I_max_ may compensate for its reduced efficacy and potency at GlyT2. It is interesting to note that it is possible to get the same overall outcome through different modulatory mechanisms, and one of the advantages of a partial GlyT2 inhibitor/GlyR PAM may be that you get the same overall outcome whilst also maintaining the capacity to recycle glycine (Figure 13).

## 5. Conclusions

The co-expression system used in this study highlights the utility of measuring the impact of GlyTs on GlyR function and also the potential for developing dual action modulators of the two proteins. It also demonstrates that compounds that are moderately effective in modulating both proteins may still be very effective in altering glycine neurotransmission. It will be of considerable interest to see how these observations translate to modulation of glycinergic neurotransmission in vivo and also how they may be applied to the treatment of pathological conditions such as chronic pain.

## Figures and Tables

**Figure 1 biomolecules-10-01618-f001:**
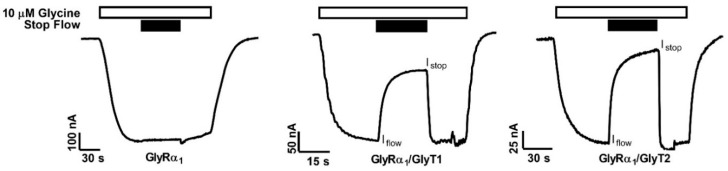
Reduction in glycine gated GlyRα_1_ currents in oocytes co-expressed with GlyTs. GlyRα_1_ is activated by 10 μM glycine (white bars). When flow of the solution is stopped on cells (black bars) expressing only GlyRα_1_ there is no change in current, however stopping the flow on oocytes co-expressing GlyRα_1_ with either GlyT2 or GlyT1 immediately causes a substantial decrease in GlyRα_1_ current amplitude. Resumption of flow restores the full amplitude of the currents. Peak current values are also decreased in co-expressed oocytes (note scales).

**Figure 2 biomolecules-10-01618-f002:**
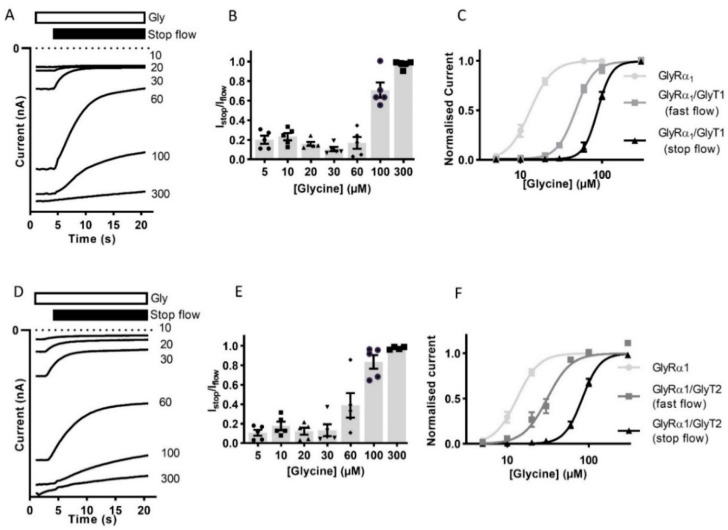
Reduction of current amplitude by GlyTs in co-expressed cells is glycine concentration dependent. Glycine concentration-dependent modulation of GlyRs by GlyTs. (**A**,**D**) Raw I_stop_ current trace examples of cells expressing GlyRα_1_/GlyT1 or GlyRα_1_/GlyT2 in the presence of varying concentrations of glycine. (**B**,**E**) I_stop_/I_flow_ ratios for varying concentrations of glycine. (**C**,**F**) Stop-flow and fast-flow (same cell) glycine dose-responses for co-expressed cells. Currents were normalised to I_max_ and fit to the Hill equation. Symbols are mean ± SEM (*n* = 5).

**Figure 3 biomolecules-10-01618-f003:**
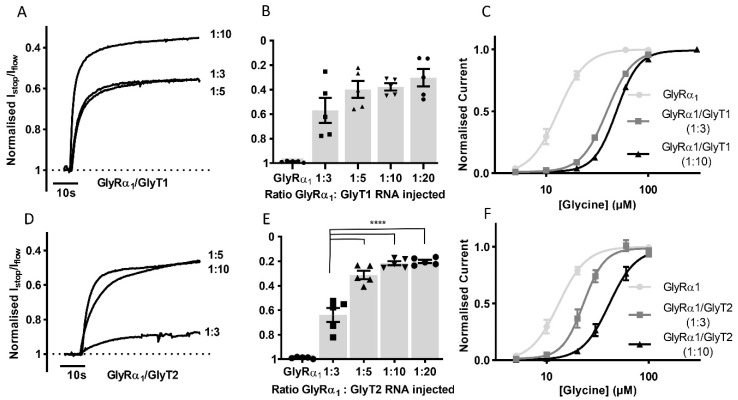
The impact of GlyT expression levels on GlyRα_1_ currents. Example traces from co-expressed oocytes expressing different (**A**) GlyT1 and (**D**) GlyT2 densities showing normalised I_stop_/I_flow_ reduction of GlyRα_1_ currents (**B**,**E**) Histograms show increasing the ratio of GlyRα_1_: GlyT cRNA injected in oocytes increases the normalised I_stop_/I_flow_ ratio when co-expressed with GlyT2 but not GlyT1. Significance was tested using a one-way ANOVA and Dunnett’s post-hoc test and **** indicates *p* < 0.0001. No significant difference was found between values for different GlyT1 ratios. (**C**,**F**) Glycine dose responses for GlyRα_1_ and GlyRα_1_/GlyT1 or GlyRα_1_/GlyT2 at 1:3 and 1:10 injected cRNA ratios. Fast flow currents were normalised to I_max_ and fit to the Hill equation. Symbols are mean ± SEM, *n* = 5.

**Figure 4 biomolecules-10-01618-f004:**
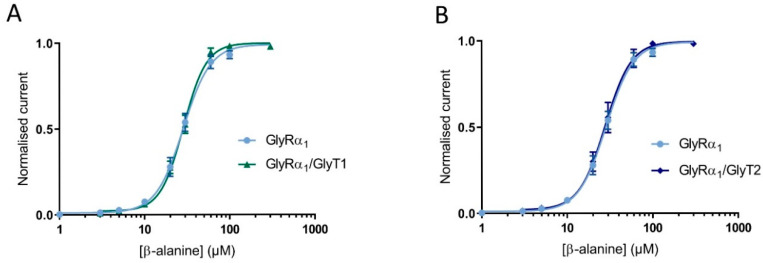
Reduction of stop-flow and fast-flow current amplitudes in co-expressed cells is dependent on GlyT transportable substrate. β-alanine dose-response curves in cells expression GlyRα_1_, GlyRα_1_/GlyT1 (**A**) or GlyRα_1_/GlyT2 (**B**) are superimposed, showing no shift in GlyRα_1_ sensitivity when a non-substrate of GlyTs is used as an agonist for GlyR. Currents were normalised to I_max_ and fit to the Hill equation. Symbols are mean ± SEM (*n* ≥ 5).

**Figure 5 biomolecules-10-01618-f005:**
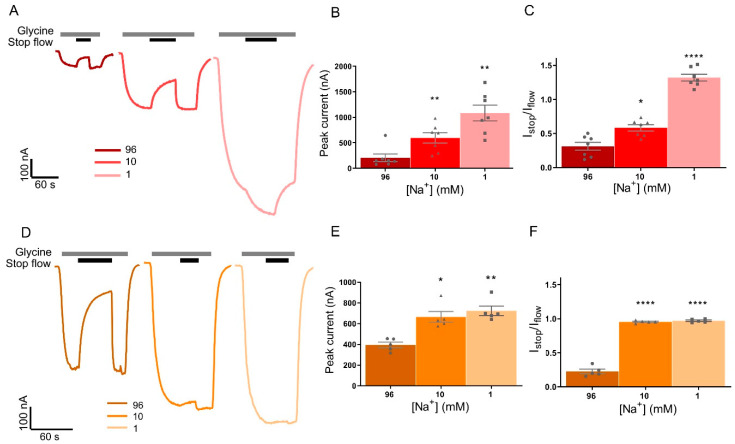
Reducing Na^+^ in the superfusing solution prevents stop-flow and fast-flow (peak) current reduction of by GlyTs. Example traces from cells co-expressing (**A**) GlyRα_1_/GlyT1 or (**D**) GlyRα_1_/GlyT2 showing stop-flow and fast-flow changes in current when 10 µM glycine (grey bars) is superfused in solutions of different Na^+^ concentrations. (**B**,**E**) Fast-flow currents are larger in low Na^+^ perfusate for both co-expressed cell types, however the absolute change is larger in GlyRα_1_/GlyT1. In GlyRα_1_/GlyT1, GlyT1 can still uptake glycine when flow is stopped on the same oocyte (black bar) in 10 mM Na^+^ solution and transport is reversed when flow is stopped (grey bar) in 1 mM Na^+^ solution (**A**, centre and right, **C**). In contrast, stop-flow uptake by GlyT2 is prevented when glycine is applied to the same GlyRα_1_/GlyT2 oocyte in 10 mM and 1 mM Na^+^ superfusing solutions (**D**, centre and right, **F**). Data is mean ± SEM, *n* ≥ 5. 10 mM and 1 mM Na^+^ values were compared to 96 mM Na^+^ values and were significance was tested using a paired *t*-test. * denotes *p* ≤ 0.05, ** denotes *p* ≤ 0.01, and **** denotes *p* ≤ 0.0001.

**Figure 6 biomolecules-10-01618-f006:**
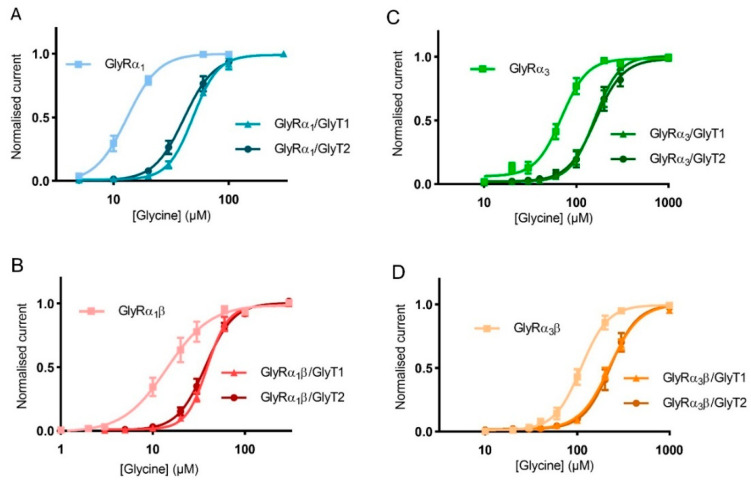
GlyTs also modulate the activity of GlyRα_3_ and GlyRα_3_β. Glycine dose responses are shifted to the right and EC_50_ values for (**A**) GlyRα_1_, (**B**) GlyRα_1_β, (**C**) GlyRα_3_ and (**D**) GlyRα_3_β are increased when co-expressed with GlyT1 or GlyT2. Currents were normalised to I_max_ and fit to the Hill equation. Symbols represent mean ± SEM, *n* = 5.

**Figure 7 biomolecules-10-01618-f007:**
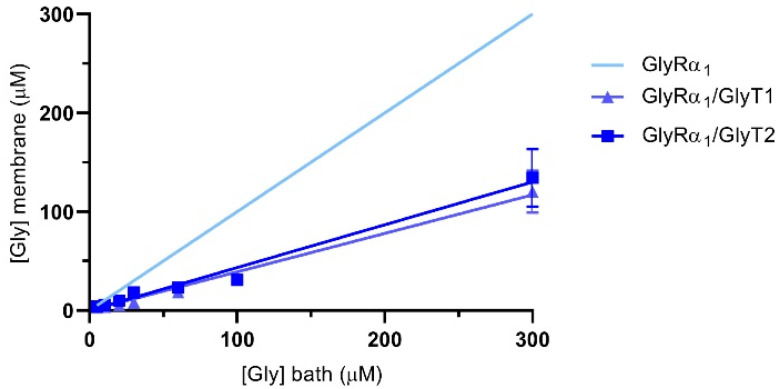
Glycine concentration sensed at the membrane of oocytes expressing GlyRα_1_/GlyTs. Since currents are reduced under fast-flow conditions, the glycine concentration at the membrane under fast-flow conditions in co-expressed oocytes was estimated by substituting EC_50_ and Hill coefficient values from cells expressing only GlyRα_1_, and current values from GlyRα_1_/GlyT1 or GlyRα_1_/GlyT2 co-expressed oocytes into the reversed Hill equation. The light blue line represents the glycine concentration sensed at the membrane when no transporters are present. Curves for GlyRα_1_/GlyT1 and GlyRα_1_/GlyT2 were fit by linear regression and values represent mean ± SEM, *n* ≥ 5.

**Figure 8 biomolecules-10-01618-f008:**
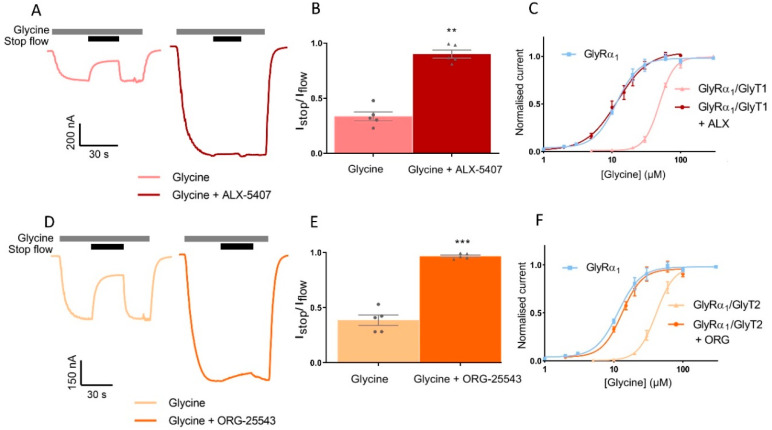
Pharmacological manipulation of stop-flow and fast-flow currents in GlyRα_1_/GlyT co-expressed cells. Example traces from cells co-expressing (**A**) GlyRα_1_/GlyT1 or (**D**) GlyRα_1_/GlyT2 showing co-application of glycine with GlyT1 inhibitor ALX-5407 or GlyT2 inhibitor ORG-25543, respectively (grey bars) prevents stop flow reduction (black bars) of glycine gated currents. Fast-flow currents are also larger when GlyT inhibitors are applied for both co-expressed cell types, suggesting GlyTs also reduce extracellularly available glycine under fast-flow conditions. I_stop_/I_flow_ values were compared between application of 10 µM glycine and 10 µM glycine in the presence of a GlyT inhibitor in the same cell expressing (**B**) GlyRα_1_/GlyT1 or (**E**) GlyRα_1_/GlyT2. ALX5-407 and ORG-25543 reduce the EC_50_s for glycine for cells co-expressing GlyRα_1_ and (**C**) GlyT1 and (**F**) GlyT2, respectively. Glycine dose-response curves for co-expressed cells in the presence of GlyT inhibitors are superimposed on dose-response curves for GlyRα_1_ alone, suggesting the effects of GlyTs in this system can be pharmacologically reversed. Currents are normalised to I_max_ and fit to the Hill equation. Data are presented as mean ± SEM (*n* ≥ 5). Values in the same cell were compared using a paired *t*-test. ** denotes *p* ≤ 0.01 and *** denotes *p* ≤ 0.001.

**Figure 9 biomolecules-10-01618-f009:**
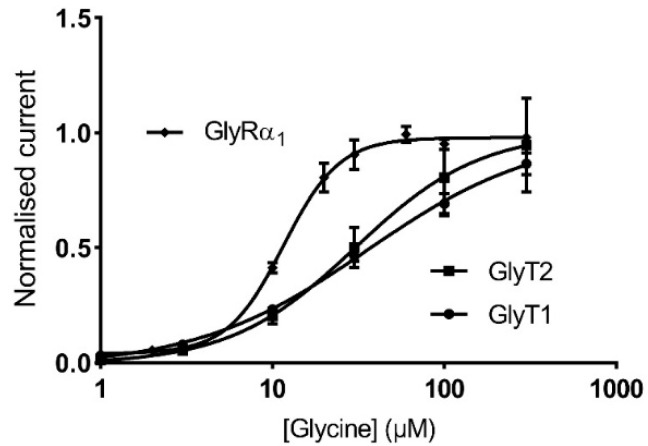
GlyTs create an apparent change in GlyRα_1_ Hill co-efficient. Glycine dose response curves for GlyRα_1_, GlyT1 and GlyT2 showing the region in which GlyTs are most effective, i.e., their EC_50_, is asymmetrical across the GlyRα_1_ curve. Currents are normalised to I_max_ and fit to the Hill equation. Symbols are mean ± SEM (*n* ≥ 5).

**Figure 10 biomolecules-10-01618-f010:**
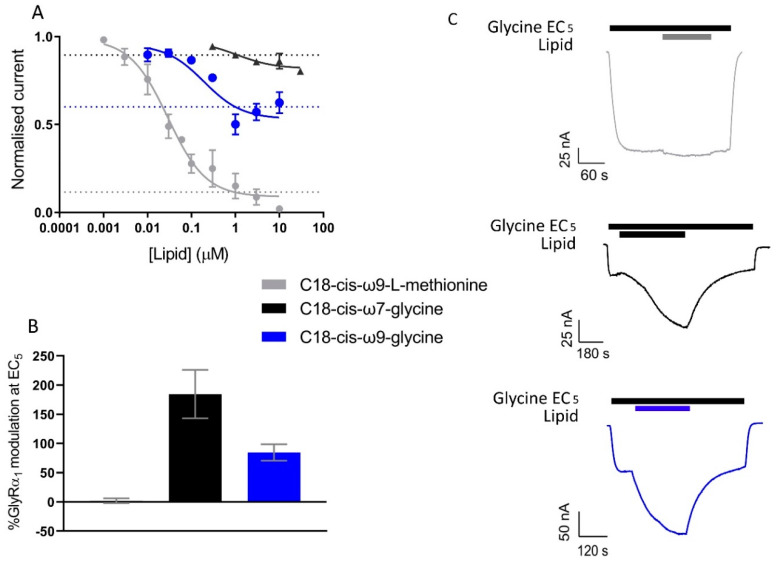
Activity of C18-*cis*-ω9-L-methionine, C18-*cis*-ω7-glycine and C18-*cis*-ω9-glycine on GlyT2 and GlyRα_1_ separately. (**A**) 30 µM glycine transport currents mediated by GlyT2 were measured in the presence of lipids in a range of concentrations. Concentration-inhibition curves for C18-*cis*-*ω*9-L-methionine, C18-*cis*-ω7-glycine and C18-*cis*-ω9-glycine. Normalised response in the presence of 1 µM lipid is marked by dashed blue, green and red lines respectively. Data from [15]. (**B**) Modulation of GlyRα_1_ currents at glycine EC_5_ by 1 µM C18-*cis*-*ω*9-L-methionine, C18-*cis*-ω7-glycine and C18-*cis*-ω9-glycine were previously measured. (**C**) Example traces of GlyRα_1_ modulation by lipids. Currents were normalised to I_max_ and fit to the Hill equation. Symbols are mean ± SEM (*n* ≥ 3). Data from [18].

**Figure 11 biomolecules-10-01618-f011:**
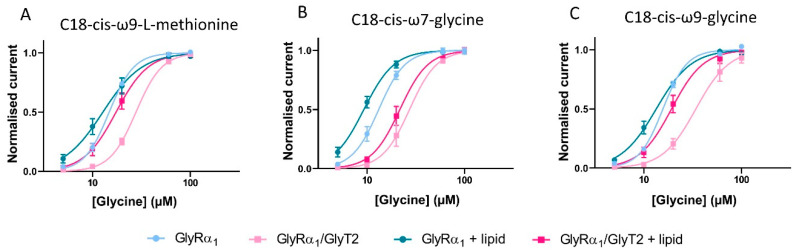
Pharmacological manipulation of fast-flow currents in GlyRα_1_ and GlyRα_1_/GlyT co-expressed cells by bioactive lipids. (**A**) C18-*cis*-*ω*9-L-methionine, (**B**) C18-*cis*-*ω*7-glycine and (**C**) C18-*cis*-*ω*9-glycine modulate the EC_50_s for glycine in cells expressing GlyRα_1_ alone and co-expressed with GlyT2. Currents were normalised to I_max_ and fit to the Hill equation. Symbols are mean ± SEM (*n* = 5).

**Figure 12 biomolecules-10-01618-f012:**
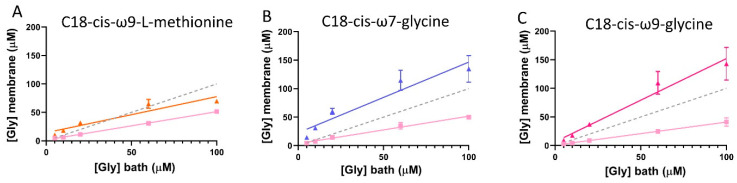
Glycine concentration sensed at the membrane of cells expressing GlyRα_1_/GlyT2 in the presence of GlyT2 specific inhibitors. Dashed line indicates reference 1:1 relationship of [Gly]_bath_ and [Gly]_membrane_. Since currents are reduced under fast-flow conditions, the glycine concentration at the membrane under fast-flow conditions in co-expressed cells was estimated by substituting EC_50_ and Hill co-efficient values from cells expressing only GlyRα_1_, and current values from GlyRα_1_/GlyT2 co-expressed cells into the Hill equation (pink lines). Similarly, the glycine concentration at the membrane under fast-flow conditions in co-expressed cells and in the presence of (**A**) the GlyT2 inhibitor, C18-*cis*-*ω*9-L-methionine (orange), (**B**) the GlyR PAM, C18-*cis*-*ω*7-glycine (purple) and (**C**) the dual action modulator C18-*cis*-*ω*9-glycine (magenta) was estimated by substituting EC_50_ and Hill co-efficient values from cells expressing GlyRα_1_/GlyT2, and current values from GlyRα_1_/GlyT2 + lipid into the Hill equation. Curves for GlyRα_1_/GlyT2 and GlyRα_1_/GlyT2 + lipid were fit by linear regression and values represent mean ± SEM, *n* ≥ 5.

**Figure 13 biomolecules-10-01618-f013:**
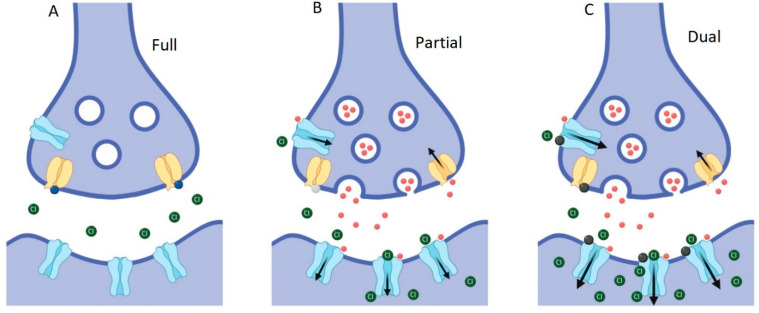
Schematic of inhibitory glycinergic synapse showing effects of modulation at GlyT2 and GlyRs. (**A**) Prolonged complete inhibition of GlyT2 (yellow) leads to steep reductions in presynaptic glycine concentrations, the lack of presynaptic vesicle filling and termination of subsequent glycine release. (**B**) Partial, non-competitive GlyT2 inhibition slows the re-uptake of glycine (red spheres), which will improve the activation of glycine gated GlyRs (blue) while still allowing refilling of vesicles to occur and subsequent glycine release into the synapse to be maintained. (**C**) Partial GlyT2 inhibition and GlyR potentiation by dual-action compounds (black spheres) will increase synaptic glycine concentrations and allow the recycling of glycine through the presynaptic terminal by GlyT2, as in B, while also directly increasing activation of GlyRs.

**Table 1 biomolecules-10-01618-t001:** Peak current reduction and stop-flow current reduction in GlyRα_1_/GlyT co-expressed cells.

	I_stop_/I_flow_	Peak Current (nA)
**GlyRα_1_**	0.98 ± 0.01	740.6 ± 105.9
**GlyRα_1_/GlyT1**	0.29 ± 0.04 ****	62.3 ± 7.9 ****
**GlyRα_1_/GlyT2**	0.23 ± 0.02 ****	150.7 ± 14.2 ****

The current values measured at I_stop_ were expressed as a fraction of currents measured at I_flow_. I_stop_/I_flow_ values and peak current values were compared between applications of 10 µM glycine in GlyRα_1_ expressing cells with GlyRα_1_ /GlyT1 or GlyRα_1_/GlyT2 expressing cells. Data are mean ± SEM (*n* = 12). Significance between values were tested using a one-way ANOVA and Dunnett’s post-hoc test and **** denotes *p* ≤ 0.0001.

**Table 2 biomolecules-10-01618-t002:** Stop-flow and fast-flow glycine EC_50_ in GlyRα_1_ and GlyRα_1_/GlyT co-expressed cells.

	Fast-Flow Glycine EC_50_ (µM)	95% CI	*n_H_*	Stop-Flow Glycine EC_50_ (µM)	95% CI	*n_H_*
**GlyRα_1_**	13.1	12.1–14.4	3.2 ± 0.4	–	–	–
**GlyRα_1_/** **GlyT1**	47.3	45.0–49.7	3.7 ± 0.2	88.3 ****	85.2–91.4	4.9 ± 0.4
**GlyRα_1_/** **GlyT2**	30.9	27.8–34.6	3.0 ± 0.47	83.1 ****	80.0–86.2	4.0 ± 0.3

Stop-flow glycine EC_50_ and Hill co-efficient (*n*_H_) values obtained from the same co-expressed cells were compared. Data are mean ± SEM or EC_50_ and 95% confidence interval (95% CI) (*n* = 5). Significance between values were tested using a paired t-test or a one-way ANOVA and Dunnett’s post-hoc test and **** denotes *p* ≤ 0.0001.

**Table 3 biomolecules-10-01618-t003:** Peak current reduction and stop-flow current reduction in GlyRα_1_/GlyT co-expressed cells in response to decreased Na^+^ concentration of buffers.

Peak Current (nA)	96 mM Na^+^	10 mM Na^+^	1 mM Na^+^
**GlyRα_1_/GlyT1**	395.2 ± 27.0	664.8 ± 52.6 *	723.2 ± 46.6 **
**GlyRα_1_/GlyT2**	204.1 ± 88.6	596.4 ± 120.8 **	1084 ± 182.6 **
**I_stop_/I_flow_**	**96 mM Na^+^**	**10 mM Na^+^**	**1 mM Na^+^**
**GlyRα_1_/GlyT1**	0.31 ± 0.06	0.58 ± 0.05 *	1.32 ± 0.06 ****
**GlyRα_1_/GlyT2**	0.23 ± 0.03	0.95 ± 0.01 ****	0.97 ± 0.01 ****

10 mM and 1 mM Na^+^ values were compared to 96 mM Na^+^ values. Significance was tested using a one-way ANOVA and Dunnett’s post-hoc test. Data is mean ± SEM, *n* ≥ 5. * denotes *p* ≤0.05, ** denotes *p* ≤ 0.01, and **** denotes *p* ≤ 0.0001.

**Table 4 biomolecules-10-01618-t004:** Fast-flow glycine EC_50_ for GlyR and GlyR/GlyT co-expressed cells using different GlyR subtypes.

	Fast-Flow Glycine EC_50_ (µM)	95% CI
**GlyRα_1_**	13.2	12.9–14.3
**GlyRα_1_/GlyT1**	48.8 ****	46.6–51.0
**GlyRα_1_/GlyT2**	41.6 ****	38.8–44.7
**GlyRα_1_β**	14.7	12.6–16.9
**GlyRα_1_β/GlyT1**	39.1 ****	37.0–41.4
**GlyRα_1_β/GlyT2**	36.8 ****	34.7–39.0
**GlyRα_3_**	64.8	59.0–70.9
**GlyRα_3_/GlyT1**	153.9 ****	143.7–164.6
**GlyRα_3_/GlyT2**	160.2 ****	145.7–175.8
**GlyRα_3_β**	109.4	102.4–117.1
**GlyRα_3_β/GlyT1**	219.7 ****	209.2–230.8
**GlyRα_3_β/GlyT2**	224.5 ****	208.6–241.5

Glycine EC_50_ values from co-expressed GlyR/GlyT cells were compared to their GlyR counterparts. Data are EC_50_ and 95% confidence interval (95% CI) (*n* ≥ 5). Significance between values were tested using a one-way ANOVA and Dunnett’s post-hoc test and **** denotes *p* ≤ 0.0001.

**Table 5 biomolecules-10-01618-t005:** Changes in I_stop_/I_flow_ and glycine EC_50_ using GlyT inhibitors in GlyRα_1_/GlyT co-expressed cells.

	I_stop_/I_flow_	Glycine EC_50_ (μM)	95% CI	*n* _H_
**GlyRα_1_**		11.2	10.0–12.4	2.2 ± 0.2
**GlyRα_1_/GlyT1**	0.39 ± 0.05	48.8 ****	46.6–51.0	3.8 ± 0.2 ***
**+ ALX-5407**	0.97 ± 0.01 **	11.3 ns	10.0–12.3	2.0 ± 0.2 ns
**GlyRα_1_/GlyT2**	0.33 ± 0.04	41.6 ****	38.8–44.7	3.2 ± 0.3 ns
**+ ORG-25543**	0.90 ± 0.04 ***	13.2 ns	11.7–14.6	2.6 ± 0.4 ns

I_stop_/I_flow_ values were compared between application of 10 µM glycine and 10 µM glycine in the presence of a GlyT inhibitor in the same cell expressing GlyRα_1_/GlyT1 or GlyRα_1_/GlyT2. Glycine EC_50_ and Hill co-efficient (*n*_H_) values for GlyRα_1_/GlyT2 and GlyRα_1_/GlyT1 in the presence and absence of GlyT inhibitors were compared to GlyRα_1_. Data are mean ± SEM or EC_50_ and 95% confidence interval (95% CI) (*n* ≥ 5). Significance between values were tested using a paired t-test or a one-way ANOVA and Dunnett’s post-hoc test. ** denotes *p* < 0.01, *** denotes *p* < 0.001, **** denotes *p* ≤ 0.0001 and ns denotes *p* > 0.05.

**Table 6 biomolecules-10-01618-t006:** Activity of C18-*cis*-ω9-L-methionine, C18-*cis*-ω7-glycine and C18-*cis*-ω9-glycine on GlyT2 and GlyRα_1_ separately.

Lipid	Structure	GlyRα_1_ Potentiation at EC_5_ (%)	GlyT2 Activity
**C18-*cis*-*ω*9-L-methionine**	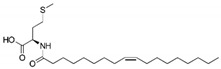	2.2 ± 4.2 (6)	**% Inhibition (1 µM)**84.9 ± 0.07 (4)**IC_50_** 29.2 nM**Max inhibition (%)** 91.2
**C18-*cis*-*ω*7-glycine**	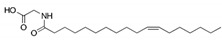	184.7 ± 41.3 (5)	**% Inhibition (1 µM)**10.5 ± 0.02 (4)**IC_50_** >10 µM**Max inhibition (%)** –
**18-*cis*-*ω*9-glycine**	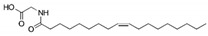	84.7 ± 14.0 (8)	**% Inhibition (1 µM)**89.6 ± 0.01 (10)**IC_50_** 31 nM**Max inhibition (%)** 95.0

Lipids were previously separately tested on cells expressing GlyT2 [15] and GlyRα_1_ [18] alone. GlyRα_1_ values indicate % stimulation of glycine EC_5_ caused by 1µM lipid. For GlyT2 activity three values are presented: the % inhibition caused by 1 µM of either lipid; the IC_50_ for inhibition of glycine transport by GlyT2; and the maximal level of inhibition observed. Values are mean ± SEM and *n* are indicated in parentheses.

**Table 7 biomolecules-10-01618-t007:** Effects of bioactive lipids on fast-flow EC_50_ and Hill co-efficient of GlyRα_1_ and GlyRα_1_/GlyT2 expressing cells.

	Glycine EC_50_ (µM)	95% CI	*n* _H_
**GlyRα_1_**	14.8	14.1–15.6	3.4 ± 0.2
**+ C18-*cis*-*ω*9-L-methionine**	12.8 ns	11.2–14.7	2.2 ± 0.3 ns
**GlyRα_1_/GlyT2**	27.9 ****	26.5–29.3	3.3 ± 0.2
**+ C18-*cis*-*ω*9-L-methionine**	17.4 *	15.7–19.5	2.6 ± 0.4 ns
**GlyRα_1_**	13.2	12.2–14.3	3.2 ± 0.3
**+ C18-*cis*-*ω*7-glycine**	9.3 ****	8.6–10.0	2.8 ± 0.3 ns
**GlyRα_1_/GlyT2**	27.3 ****	23.6–31.4	3.1 ± 0.5
**+ C18-*cis*-*ω*7-glycine**	21.7 ****	19.7–24.4	3.0 ± 0.5 ns
**GlyRα_1_**	15.7	15.1–16.4	3.5 ± 0.2
**+ C18-*cis*-*ω*9-glycine**	13.4 ns	12.4–14.5	2.4 ± 0.2 ns
**GlyRα_1_/GlyT2**	34.1 ****	29.6–39.4	2.6 ± 0.3
**+ C18-*cis*-*ω*9-glycine**	18.0 *	16.1–20.2	2.7 ± 0.4 ns

Glycine EC_50_ and Hill co-efficient (*n*_H_) values for GlyRα_1_/GlyT2 and GlyRα_1_/GlyT1 in the presence and absence of lipid were compared to GlyRα_1_. Data presented are EC_50_ and 95% confidence interval (95% CI). Hill co-efficient (*n*_H_) values for GlyRα_1_ ± lipid or GlyRα_1_/GlyT2 ± lipid were compared. Data are mean ± SEM (*n* = 5). Significance between values were tested using a one-way ANOVA and Dunnett’s post-hoc test. * denotes *p* ≤ 0.05, **** denotes *p* ≤ 0.0001 and ns denotes *p* > 0.05.

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
