# Peer review of "A System for Assessing Dual Action Modulators of Glycine Transporters and Glycine Receptors"

_biomolecules, 2020, doi:10.3390/biom10121618_

Round 1

Reviewer 1 Report

In the manuscript “A System for Assessing Dual Action Modulators of Glycine Transporters and Glycine Receptors” by Sheipouri et al., that was submitted for publication to biomolecules, the authors describe a novel xenopus laevis based coexpression system that allows to analyse the functional interactions of glycine receptors and transporters. The authors demonstrate that upon coexpression of each of the GlyTs together with glycine receptors can limit the local availablility of glycine at the receptor especially when low or intermediate glycine concentrations are applied, and this effect was dependent on the activity of the respective transporter and their expression level. They subsequently used this system to monitor the effects of different glycine/lipid conjugates that have been previously characterized by this group to act specifically on the GlyT2, GlyRs or both. Here they could show, that substances that inhibit GlyT2 and in addition work as a positive allosteric modulator of the GlyR shows strongest effects on the total glycine induced current in this co-expression system.

In general, the study is well conceived, and the experiments support the conclusions reached by the authors. The main findings, how (and at which concentration range) the GlyTs are able to contribute to the regulation of glycinergic currents, are novel and interesting for a broader readership, therefore justifying publication in Biomolecules. There are, however, some points that require clarification before publication.

Specific points:

Stop/flow experiments: The authors demonstrate that in oocytes coexpressing a GlyT1 and GlyR, a significant reduction in the observed glycine induced current occurs after stopping of the superfusion solution. Does this also occur in oocytes only expressing a GlyT?

Furthermore the authors claim, but do not show that the currents observed in their stop flow experiments are almost exclusively mediated by the GlyRs. The data in Fig. S2 however suggest that especially at low concentration (i.e. 10 µM, which is used for some of the experiments 50 % of the current for GlyT1/GlyR coexpressing oocytes were mediated by GlyT1, whereas still 20 % of the current was transporter carried in GlyT2/GlyR expressing oocytes). So, are the currents seen e.g. in Fig. 1 GlyR, GlyT or mixed currents?

How do the authors interprete the difference in the contribution of the individual transporters to the total current? Is this due to differences in expression level of a results e.g. from differences in the transport kinetics?

Figure 3: Here, although I understood the general finding, I found the figure rather confusing:

The authors claim that the more GlyT is present, the stronger the reduction in the current observed after stopping of the superfusion. In A and D: Why is the ratio between the Istop/Iflow is close to 0 at the beginning of the experiment? As far as I understood, the current is expected to be maximal at this point and should decrease from there on? In B and E the ratio of the Istop/Iflow is depicted. This should be, following the authors description be around 1 for oocytes expressing only GlyR, but it is close to 0… so what is actually depicted in this graph?

Figure 5: How do the authors interprete the apparent facilitation of the glycine induced current under the stopped flow conditions? Is this really elicited by glycine released be the transporter? Is fo can this be blocked by GlyT1 inhibitors? Why is this not seen in GlyT2 expressing oocytes, since the major driving force is Na+, I would expect the observed effect to be even stronger?

General: The glycine affinity the authors determine for essentially all GlyR subunits is very high (in the area of roughly 10 µM)! on most studiy using a comparable oocyte expression system much lower affinities ( in the range of 100-300 µM) were described. The authors should at least discuss possible reasons for these differences.

The solution exchange system the authors are using is very slow. This has of course a least some impact both on the overall shape of the observed current traces. Additionally, however, slow developing changes in the local concentrations of the ions, especially chloride, cannot be excluded. This might affect both the observed overall current and the activity of the GlyTs. Can the authors at least comment on this possibility?

The authors describe nicely that in the coexpression system a marked change in the Hill coefficient was observed. If the only effect of the coexpressed GlyT is as the authors suggest a modulation of the local glycine concentration such a change in the Hill coefficient is not to be expected. So, how do the authors explain this effect?

Reviewer 2 Report

Review of manuscript „A System for Assessing Dual Action Modulators of Glycine Transporters and Glycine Receptors”

The manuscript describes the evaluation of dual action modulators of both GlyTs

and GlyRs in the treatment of ailments associated with impaired glycinergic neurotransmission. The topic of the paper appears to be interesting, the authors did several minor mistakes which are listed below.

  • The authors should highlight the relevance of their studies in the introduction.
  • The conclusion in abstract is insufficient.
  • Please remove double spaces in the manuscript.
  • Materials and methods: Please provide the numer of ethical approval.
  • What are the other agonists of GlyRs? The authors could mention them, at least.
  • The authors should also estimate the expression at mRNa and protein level of both GlyR and GlyT.
  • Please provide the structure of ALX-5407

Round 2

Reviewer 1 Report

From my Point of view all Points raised in the initial review have been addressed appropriately.